# MT-Video-Bench: A Holistic Video Understanding Benchmark for Evaluating Multimodal LLMs in Multi-Turn Dialogues

## Abstract

The recent development of Multimodal Large Language Models (MLLMs) has significantly advanced AI's ability to understand visual modalities. However, existing evaluation benchmarks remain limited to single-turn question answering, overlooking the complexity of multi-turn dialogues in real-world scenarios. To bridge this gap, we introduce **MT-Video-Bench**, a holistic video understanding benchmark for evaluating MLLMs in multi-turn dialogues. Specifically, our MT-Video-Bench mainly assesses six core competencies that focus on perceptivity and interactivity, encompassing 987 meticulously curated multi-turn dialogues from diverse domains. These capabilities are rigorously aligned with real-world applications, such as interactive sports analysis and multi-turn video-based intelligent tutoring. With MT-Video-Bench, we extensively evaluate various state-of-the-art open-source and closed-source MLLMs, revealing their significant performance discrepancies and limitations in handling multi-turn video dialogues. The benchmark will be publicly available to foster future research.

## 1 Introduction

The rapid progress of Multimodal Large Language Models (MLLMs) has markedly advanced AI's capacity to perceive and reason over visual modalities, especially when integrated with natural language. Recent systems such as Qwen2.5-VL (Bai et al., 2025), InternVL3.5 (Wang et al., 2025b), and Gemini 2.5 (Team, 2025) demonstrate impressive performance in single-turn video question answering and long-form video comprehension (Zhang et al., 2023; Rawal et al., 2024; Sun et al., 2022; Wang et al., 2024c; Chandrasegaran et al., 2024). Yet, real-world human–AI interaction is rarely confined to single-turn queries. Instead, it typically unfolds as multi-turn dialogues, where users iteratively refine their questions, shift topics, and expect contextually coherent responses grounded in video content. This interactive setting poses unique challenges: models must not only recall and integrate prior dialogue history but also adapt to conversational dynamics, such as handling topic shifting or gracefully refusing unanswerable queries.

Despite these demands, existing video understanding benchmarks (Fu et al., 2025; Wang et al., 2024b; Zhou et al., 2025; Ma et al., 2025) predominantly focus on single-turn evaluation, emphasizing factual perception of video content—such as recognizing objects, actions, or temporal relations—while neglecting dialogue-level reasoning. A few recent efforts explore long-context or multi-shot video benchmarks, yet they fall short of capturing the interplay between perceptivity (faithfully interpreting multimodal input) and interactivity (sustaining natural, user-aware conversations). Consequently, **the community lacks a rigorous and holistic framework to measure how well MLLMs can operate in realistic multi-turn, video-grounded dialogues.**

To fill this gap, as shown in Figure 2, we introduce **MT-Video-Bench**, a holistic benchmark for evaluating MLLMs in multi-turn video dialogue. MT-Video-Bench systematically targets six core capabilities spanning perceptivity (object reference, memory recall, and content summary) and interactivity (answer refusal, topic shifting, and proactive interaction). The benchmark comprises 987 carefully curated dialogues across 135 videos, covering diverse domains such as sports, education, and daily activities. Moreover, unlike prior datasets, MT-Video-Bench emphasizes cross-scene

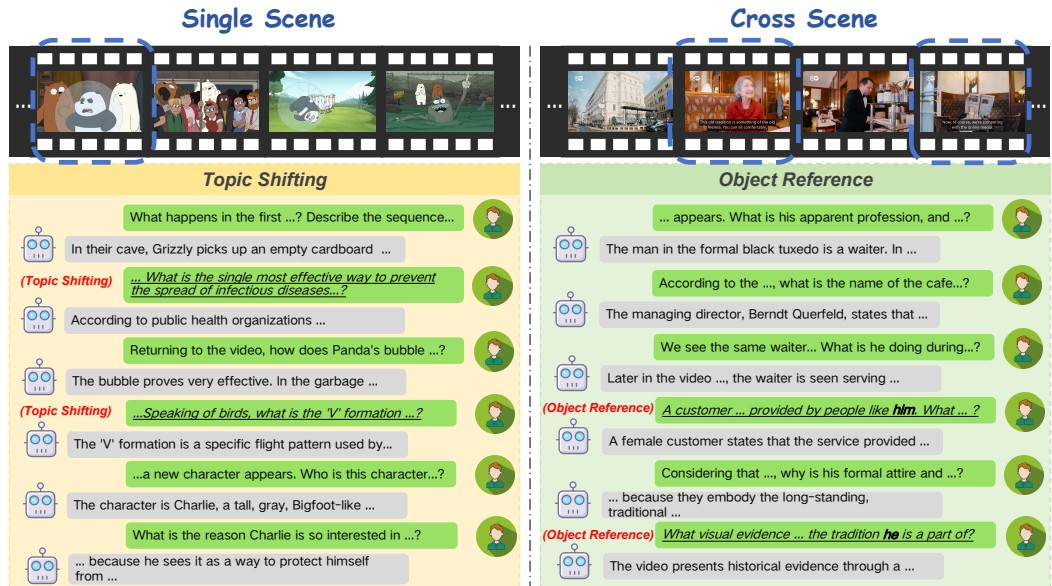

Figure 1: Illustration of multi-turn dialogues under single-scene and cross-scene settings. The evaluated questions corresponding to tasks are marked with underlining, and the scenes involved in the entire multi-turn dialogues are marked with blue dotted boxes.

reasoning, long-range dependencies, and interactive adaptability, thereby aligning closely with real-world application demands.

Based on our MT-Video-Bench, we provide a detailed evaluation of both open-source and closed-source models, highlighting the current limitations and performance discrepancies in different abilities. Specifically, several insightful findings are as follows:

- The perceptual and interactive capabilities of MLLMs in multi-turn dialogues still have significant room for improvement. On MT-Video-Bench, even the strongest closed-source model Gemini 2.5 Pro achieves only 68.45% overall accuracy, while most open-sourced MLLMs exhibit accuracies below 50%, except for the Qwen2.5-VL and InternVL3.5 series.

- Performance is imbalanced across different tasks and scene types. MLLMs generally perform better on perceptual subtasks (e.g., Object Reference) than on interactive ones (e.g., Proactive Interaction), with a substantial gap between closed- and open-source models. Moreover, all models tend to perform worse in cross-scene settings compared to single-scene tasks.

- Model scaling is beneficial but not sufficient. Larger models consistently outperform smaller counterparts across most subtasks, yet scaling alone does not ensure consistent improvements. For example, in the InternVL 3.5 series, enabling the Thinking mode allows smaller models to achieve performance comparable to that of larger models, which demonstrates the significant benefit of the reasoning process in enhancing model performance.

To summarize, the contributions of this paper are as follows: We identify the critical gap in evaluating multi-turn video-grounded dialogues and propose the MT-Video-Bench, the first holistic benchmark that operationalizes this evaluation via six well-defined capabilities across 987 dialogues and 5,805 QA pairs. Then, based on extensive experiments on MT-Video-Bench, we underscore the challenges and potential directions for improvement of handling and reasoning over multi-turn dialogues, offering a roadmap for future research and development.

## 2 RELATED WORK

**Multimodal LLMs.** MLLMs have become a central research focus in advancing general-purpose intelligence. By jointly modeling textual and visual modalities, these models are able to capture

Table 1: Comparison with other benchmarks. **Avg. Q/V**: the average number of QA pairs per video. **Long**: whether the average video length is greater than 10 minutes. **Cross-Scene**: whether the dialogue covers more than 4 scenes.

| Benchmark | #QAs | Avg. Q/V | Long | Dialogue | #Turns | Cross-Scene | Annotation |
|---|---|---|---|---|---|---|---|
| MVBench (Li et al., 2024c) | 4,000 | 1.00 | ✗ | ✗ | 1.00 | - | Auto |
| LongVideoBench (Wu et al., 2024b) | 6,678 | 1.77 | ✗ | ✗ | 1.00 | - | Manual |
| Video-MME (Fu et al., 2025) | 2,700 | 3.00 | ✔ | ✗ | 1.00 | - | Manual |
| LVBENCH (Wang et al., 2024b) | 1,549 | 15.04 | ✔ | ✗ | 1.00 | - | Manual |
| MLVU (Zhou et al., 2025) | 3,102 | 1.79 | ✔ | ✗ | 1.00 | - | Manual |
| Video-MMLU (Song et al., 2025) | 15,746 | 14.78 | ✗ | ✗ | 1.00 | - | Auto&Manual |
| ScaleLong (Ma et al., 2025) | 1,747 | 6.49 | ✔ | ✗ | 1.00 | - | Manual |
| SVBench (Yang et al., 2025) | 7,374 | 36.87 | ✗ | ✔ | 4.29 | ✗ | Auto&Manual |
| **MT-Video-Bench (Ours)** | 5,805 | 43.00 | ✔ | ✔ | 5.88 | ✔ | Auto&Manual |

cross-modal dependencies and enhance semantic reasoning(Zhu et al., 2023; Ma et al., 2024; Zhang et al., 2024a; Wang et al., 2025a; 2024a). Recent advances have further extended MLLMs to the video domain, enabling video understanding, which subsequently supports dialogue(Li et al., 2023; Cheng et al., 2024; Maaz et al., 2023). For example, Qwen2.5-VL (Bai et al., 2025) employs a dynamic-resolution Vision Transformer with MRoPE for spatiotemporal alignment, and connects an MLP merger to the Qwen2.5 LLM decoder. InternVL3.5 (Wang et al., 2025b) integrates InternViT as the vision encoder with a ViT-MLP-LLM paradigm, and further adopts Visual Resolution Router (ViR) with Visual Consistency Learning (ViCO) for cross-modal alignment.

**Video Benchmarks**. Significant developments have also been made in video understanding benchmarks(Wang et al., 2023; Wu et al., 2024a; Xiao et al., 2021). For example, MVBench (Li et al., 2024c) focuses on concise video QA tasks to evaluate multimodal understanding abilities, while MLVU (Zhou et al., 2025) and LVBENCH (Wang et al., 2024b) provide a comprehensive analysis for MLLMs' long-video understanding performance. MMBench-Video (Fang et al., 2024) is a long-form, multi-shot benchmark that evaluates fine-grained abilities of MLLMs, including temporal reasoning, perception, and general reasoning in video understanding. SVBench (Yang et al., 2025) is a benchmark for temporal multi-turn dialogues in streaming videos, designed to assess the capabilities of streaming video understanding of MLLMs. However, prior benchmarks primarily focus on evaluating the video understanding capabilities of models, overlooking the multi-turn dialogue capabilities, which require not only the ability to recall contextual information but also to engage in coherent, interactive communication with users across multiple turns.

## 3 MT-VIDEO-BENCH

### 3.1 OVERVIEW

MT-Video-Bench is designed to comprehensively evaluate the "Perceptivity" and "Interactivity" of MLLMs in multi-turn video-grounded dialogues. Different from conventional video understanding benchmarks that primarily focus on single-turn question answering, MT-Video-Bench is specifically designed to mimic real-world interactive scenarios, emphasizing contextual coherence, cross-scene video comprehension, and adaptive interactivity.

MT-Video-Bench systematically evaluates six core capabilities of MLLMs through 987 meticulously curated multi-turn dialogues with 5,805 QA pairs. Each conversation requires not only accurate video perception but also contextual reasoning within or across video scenes, with representative examples shown in Figure 1.

A comprehensive comparison between our MT-Video-Bench and other related benchmarks is provided in Table 1. MT-Video-Bench presents the following critical values: (1) supports multi-turn dialogues that evaluate contextual coherence and long-range dependency, (2) supports cross-scene reasoning that requires integrating information across different video clips, and (3) provides a fine-grained assessment of perceptivity and interactivity through six tasks.

Figure 2: An overview of the semi-automatic data construction process of MT-Video-Bench.

## 3.2 EVALUATION TASKS

**Perceptivity** assesses the model's foundational ability to perceive and integrate information from both the visual video content and the multi-turn conversational context. This capability is essential for accurately understanding user queries and generating contextually grounded responses throughout the dialogue. It includes:

- **Object Reference (OR)** evaluates the model's ability to resolve references and pronouns in the user's input, ensuring that entities mentioned implicitly are correctly mapped to the appropriate objects, characters, or concepts.

- **Memory Recall (MR)** measures the model's capacity to retrieve, retain, and integrate relevant information from prior conversational turns or long-term history, enabling coherent reasoning and continuity across interactions.

- **Content Summary (CS)** assesses the model's effectiveness in condensing conversational and video content into succinct yet comprehensive summaries, while preserving essential details, coherent structure, and semantic fidelity.

**Interactivity** evaluates the model's capacity to conduct coherent, adaptive, and user-aware dialogues based on the video content. It focuses on appropriately refusing unanswerable questions, smoothly adapting to topic changes, and proactively maintaining engagement. It includes:

- **Answer Refusal (AR)** tests the ability to recognize unanswerable queries based on available evidence and explicitly decline or indicate insufficiency without hallucination.

- **Topic Shifting (TS)** evaluates how effectively the model can track and adapt to user-initiated changes in conversational focus or subject matter, while maintaining coherence, fluency, and relevance throughout the dialogue.

- **Proactive Interaction (PI)** probes the model's capacity to sustain or restore engagement through clarifications, elaborations, or novel insights when signs of disinterest or disengagement are detected, thereby fostering renewed interest and continuation of the dialogue.

## 3.3 DATA COLLECTION

As shown in Figure 2, the data collection process for MT-Video-Bench involves both automated construction and human verification. We first acquire videos from online platforms and split them into single-scene segments. Next, we retrieve and merge relevant scenes by extracting frames, performing object detection, and constructing object memory bank. Multi-turn dialogues are then generated automatically for diverse evaluation tasks. Finally, human annotators are involved to ensure the accuracy and quality of the generated dialogues.

**Video Collection and Single-Scene Splitting.** The data collection process begins with the manual acquisition of 135 videos from various online platforms, such as YouTube, within the past year.

Subsequently, we employ PySceneDetect[1] to divide the videos into shorter clips. Recognizing that these clips are often too brief to represent complete scenes, we then use the Gemini 2.5 Flash model (Team, 2025) to generate descriptive captions for each clip. Finally, the caption-based clip merging method is iteratively applied twice to combine related clips into a coherent, single-scene video, ensuring a seamless and contextually accurate representation of the scene. These refined single-scene videos serve as the core visual content for the subsequent task of generating single-scene, multi-turn dialogues.

**Cross-Scene Video Merging.** The generation of cross-scene, multi-turn dialogues necessitates the retrieval and merging of relevant scenes from disparate video segments, which serves as a critical step in creating coherent interactions that span across multiple visual contexts. Firstly, frames are extracted from the video at 2 FPS and then filtered based on two criteria: sharpness and similarity to the previous selected frame. The sharpness of each frame is evaluated by the Laplace Operator to ensure that only clear, visually significant frames are retained, improving the overall quality of the selected frames. To avoid redundancy, frames with high similarity to the preceding selected frame are discarded. Specifically, a histogram-based image similarity calculation method is used to compare consecutive frames, excluding those with a similarity score above 0.9. This approach ensures that the selected frames are distinct and capture key moments in the video.

Following frame selection, object detection is performed using YOLOv11 (Khanam & Hussain, 2024), and each detected object is then annotated with a caption generated by the Gemini 2.5 Flash (Team, 2025), providing detailed descriptions for each object. As the video progresses, a dynamic object memory bank is maintained, continuously expanded based on object captions and visual similarities. This memory bank associates unique object IDs with their corresponding attributes, enabling the identification of the same objects across frames. To merge relevant scenes, a retrieval step across scenes is performed to select video segments that share common objects or themes, which are then merged to ensure continuity both thematically and contextually.

**Multi-Turn Dialogues Generation.** This process employs the Gemini 2.5 Pro (Team, 2025) to automate the generation of both single-scene and cross-scene multi-turn dialogues, based on the six evaluation tasks defined earlier. For each video, we generate multiple multi-turn dialogues, each corresponding to different scenes. To determine the most appropriate task for each scene, we prompt MLLMs to evaluate the scene's capabilities, scoring them on a scale from 1 to 6. Only those tasks that receive a score of 5 or 6 are selected for dialogue generation. For multi-turn dialogues spanning multiple scenes, we specifically adopt an object-centered approach for cross-scene question design since objects often serve as the central element around which events unfold. This approach emphasizes the continuity and relationships of objects across scenes, enabling the generation of dialogues that are both contextually consistent and thematically coherent.

## 3.4 QUALITY CONTROL

Following automated data collection, we employ the following two-stage human verification process to enhance dataset quality.

**Stage 1: Eliminating information leakage.** We categorize all benchmark questions into two types: (1) context-dependent, which can be answered solely based on dialogue history; and (2) video-dependent, which require direct grounding in the video content. We observe that in some generated dialogues, earlier QA pairs embedded excessive background hints, enabling models to answer subsequent questions without relying on the video. This led to an overrepresentation of context-dependent items, thereby weakening the evaluation of video understanding. To mitigate this, we systematically removed such cases to ensure that the majority of questions required genuine video-based reasoning.

**Stage 2: Human verification and validation.** After the first filtering, human annotators conducted a secondary review from a human perspective. We verify whether each question and answer pair was factually aligned with the video and free from ambiguities. Beyond factual correctness, we also examine whether each question is properly aligned with its intended ability dimension. For example, answer refusal questions must explicitly test whether a model can recognize "events absent from the video," while object reference questions must involve pronoun disambiguation. Any misaligned

---

[1]https://github.com/Breakthrough/PySceneDetect

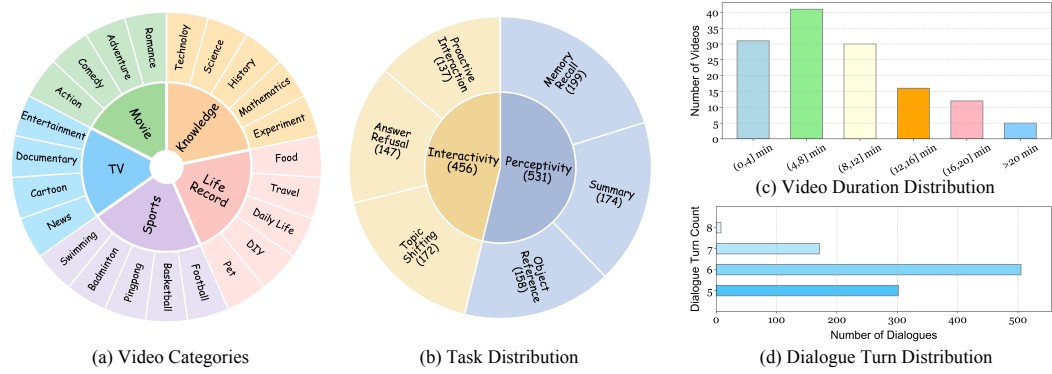

(a) Video Categories      (b) Task Distribution      (c) Video Duration Distribution

                                                    (d) Dialogue Turn Distribution

Figure 3: Overview of MT-Video-Bench. (a) Video Categories. MT-Video-Bench includes videos spanning 5 major categories, ensuring diverse topical coverage. (b) Task Distribution. MT-Video-Bench consists of a total of 6 tasks with a relatively balanced distribution. (c) Video Duration Distribution. MT-Video-Bench includes both long and short videos. (d) Dialogue Turn Distribution. Multi-turn dialogues in MT-Video-Bench involve 5 to 8 rounds.

samples are discarded. Finally, we filtere out overly simple questions, as they can be trivially solved by most models and fail to highlight multi-turn reasoning and video comprehension capacities.

## 3.5 DATASET STATISTICS

Figure 3 presents the statistics of MT-Video-Bench. It covers a broad range of topics across five main categories: Movie, TV, Sports, Knowledge, and Life Record, each with multiple sub-topics, ensuring a diverse and balanced data distribution. With a total of 987 multi-turn dialogues, the data distribution across the six primary tasks in MT-Video-Bench is relatively balanced, as shown in Figure 3 (b). Furthermore, our dataset features videos of varying lengths, with most being under 15 minutes and a small proportion exceeding 15 minutes, thereby ensuring coverage of both short and long videos. The number of dialogue turns typically ranges from 5 to 8, with an average of 5.88 turns per dialogue.

## 3.6 EVALUATION METHOD

In multi-turn dialogues, each new turn depends on the interactions between users and assistants in previous turns. This dynamic is particularly crucial in tasks that involve high interactivity, such as proactive interactions. Therefore, we follow the multi-turn dialogue evaluation setup used in LLMs (Bai et al., 2024), leveraging our meticulously curated dataset as the golden context for dialogue history, rather than relying on self-predicted context from MLLMs.

For evaluation, we first use Gemini 2.5 Flash (Team, 2025) to construct a checklist for each QA pair. Specifically, each checklist consists of five yes/no questions designed to assess the accuracy of the model's responses and its performance on specific tasks. Then, manual validation is employed to filter out unqualified checklists. After filtering, each QA pair has an average of 3.29 questions in the final checklists, with 62.35% answered as yes and 37.65% as no. During the evaluation process, Gemini 2.5 Flash (Team, 2025) is used to answer each checklist question based on the model-generated answers. The evaluation metric is calculated as the accuracy (ACC), based on the proportion of correct answers across all checklists.

## 4 EXPERIMENTS

### 4.1 EXPERIMENTAL SETTINGS

For closed-source models, we evaluate Gemini 2.5 Pro (Team, 2025), Gemini 2.5 Flash (Team, 2025), and Doubao-Seed-1.6-vision (Seed, 2025). For open-source models, we select 18 representative MLLMs, including Qwen2.5 VL series (Bai et al., 2025), InternVL3.5 series (Wang et al.,

Table 2: Evaluation results on MT-Video-Bench. **OR**: Object Reference. **MR**: Memory Recall. **CS**: Content Summary. **AR**: Answer Refusal. **TS**: Topic Shifting. **PI**: Proactive Interaction. The best performance and the second best performance are highlighted in green and blue, respectively.

| Models | Overall | Perceptivity | | | Interactivity | | |
|---|---|---|---|---|---|---|---|
| | | OR | MR | CS | AR | TS | PI |
| *Closed-Sourced Models* | | | | | | | |
| Gemini 2.5 Pro (Team, 2025) | 68.45 | 66.13 | 67.80 | 80.49 | 67.50 | 73.67 | 55.12 |
| Gemini 2.5 Flash (Team, 2025) | 63.30 | 63.44 | 63.41 | 73.48 | 64.32 | 68.12 | 47.04 |
| Doubao-Seed-1.6-vision (Seed, 2025) | 58.55 | 66.19 | 60.85 | 68.95 | 43.84 | 65.99 | 45.50 |
| *Open-Sourced Models* | | | | | | | |
| *Model Size > 8B* | | | | | | | |
| Qwen2.5-VL-72B (Bai et al., 2025) | 58.48 | 60.60 | 56.40 | 74.20 | 57.07 | 64.27 | 38.35 |
| InternVL3.5-38B (Think) (Wang et al., 2025c) | 58.11 | 60.87 | 60.36 | 69.90 | 46.86 | 65.17 | 45.51 |
| Qwen2.5-VL-32B (Bai et al., 2025) | 57.88 | 60.20 | 59.63 | 74.88 | 50.71 | 63.41 | 38.47 |
| InternVL3.5-38B (No Think) (Wang et al., 2025c) | 50.04 | 52.51 | 46.37 | 61.86 | 44.24 | 58.78 | 36.46 |
| *4B < Model Size ≤ 8B* | | | | | | | |
| InternVL3.5-8B (Think) (Wang et al., 2025c) | 56.29 | 57.81 | 54.82 | 73.18 | 47.62 | 62.50 | 41.84 |
| Qwen2.5-VL-7B (Bai et al., 2025) | 53.12 | 56.18 | 49.99 | 67.21 | 52.20 | 57.20 | 35.92 |
| InternVL3.5-8B (No Think) (Wang et al., 2025c) | 49.35 | 51.71 | 46.95 | 61.50 | 40.83 | 57.23 | 37.85 |
| LLaVA-Video-7B (Zhang et al., 2025b) | 49.17 | 53.85 | 43.57 | 63.64 | 41.32 | 56.67 | 35.98 |
| MiniCPM-o (Yao et al., 2024) | 48.41 | 55.06 | 43.27 | 61.59 | 34.58 | 57.53 | 38.43 |
| MiniCPM-V4.5 (Yao et al., 2024) | 47.06 | 51.57 | 43.08 | 56.17 | 38.46 | 52.58 | 40.47 |
| InternVideo2.5-8B (Wang et al., 2025e) | 47.04 | 44.87 | 43.49 | 60.33 | 45.23 | 54.81 | 33.50 |
| VideoLLaMA3-7B (Bai et al., 2025) | 46.06 | 52.06 | 42.40 | 55.74 | 45.23 | 48.25 | 32.69 |
| LLaVA-OneVision-7B (Li et al., 2024a) | 45.75 | 50.01 | 43.36 | 59.34 | 32.79 | 55.44 | 33.56 |
| VideoChat-Flash-7B (Li et al., 2024d) | 41.11 | 47.92 | 39.33 | 51.14 | 28.02 | 48.27 | 32.01 |
| LLaVA-NeXT-Video-7B (Zhang et al., 2024c) | 38.04 | 43.05 | 36.04 | 48.58 | 27.60 | 42.94 | 30.00 |
| *Model Size ≤ 4B* | | | | | | | |
| InternVL3.5-4B (Think) (Wang et al., 2025c) | 52.25 | 54.94 | 53.78 | 67.50 | 37.74 | 54.67 | 44.89 |
| Qwen2.5-VL-3B (Bai et al., 2025) | 48.07 | 50.64 | 43.54 | 65.82 | 46.80 | 50.33 | 31.30 |
| InternVL3.5-4B (No Think) (Wang et al., 2025c) | 45.90 | 46.03 | 46.19 | 61.30 | 30.41 | 55.72 | 35.74 |

2025c), LLaVA-Onevision series (Li et al., 2024b), InterVideo2.5 series (Wang et al., 2025d), LLaVA-Video series (Zhang et al., 2024d), LLaVA-NeXT-Video series (Zhang et al., 2024b), VideoChat-Flash series (Li et al., 2024e), VideoLlama3 series (Zhang et al., 2025a) and MiniCPM series (Yao et al., 2024).

**Evaluation.** For each model, we adopt a uniform sampling strategy to process video frames, setting the number of frames to 32. Each video is resized before input to models that the longer side is limited to 720 pixels and the other side is scaled proportionally. More details are described in Appendix C.1. For the prompts, we provide the evaluation prompts of six tasks of MT-Video-Bench in C.2.

## 4.2 MAIN RESULTS

As shown in Table 2, we provide the performance results of different MLLMs on our MT-Video-Bench, and we have the following insightful and interesting observations:

- MT-Video-Bench is very challenging. Even the best-performing closed-source model, Gemini 2.5 Pro, only achieves 68.45% overall accuracy, which is inferior to the performance of human experts a lot.

- Among all evaluated models, Gemini 2.5 Pro consistently ranks first in both overall accuracy and every individual subtask. While closed-source systems still dominate overall performance, some open-source models demonstrate competitive results in specific dimensions. For example, Qwen2.5-VL-72B shows strong ability in MR, narrowing the gap with Gemini 2.5 Pro. However, on interaction-related subtasks such as AR, the performance difference between open-source and closed-source models remains substantial.

- Results vary significantly across different dimensions, and models generally perform better on perception-related subtasks, where large-scale models generally achieve stronger socres, sometimes exceeding 60. For example, the average score of OR is 54.55, while for PI is 38.60.

- Larger models tend to achieve higher accuracy. For instance, within the Qwen2.5-VL series, the 72B and 32B models significantly outperform the 7B and 3B variants across nearly all subtasks. Similarly, larger InternVL3.5 models achieve better results than their smaller counterparts. However, sometimes small MLLMs can lead to higher scores. For instance, the AR scores for Qwen2.5-VL-7B, Qwen2.5-VL-32B, and InternVideo2.5-8B are 52.20, 50.71, and 45.23, respectively. In addition, enabling *thinking mode* within the same model variant leads to significant performance improvements, suggesting that inference strategies, beyond model size, can substantially affect benchmark outcomes.

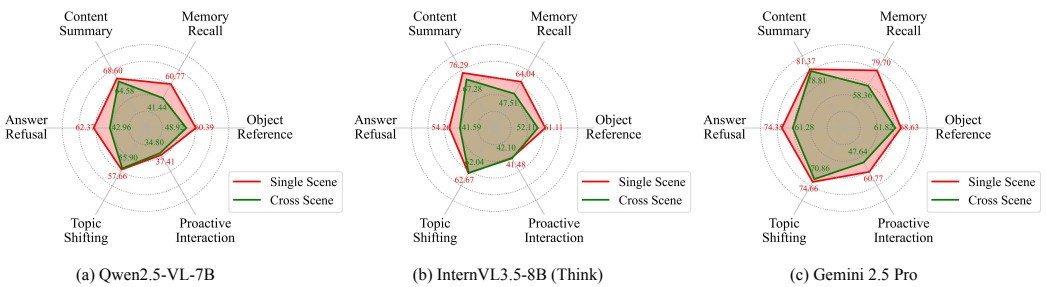

(a) Qwen2.5-VL-7B      (b) InternVL3.5-8B (Think)      (c) Gemini 2.5 Pro

Figure 4: Performance comparison of Qwen2.5-VL-7B, InternVL3.5-8B (Think), and Gemini 2.5 Pro across various tasks under single-scene and cross-scene settings.

### 4.3 FURTHER ANALYSIS

#### 4.3.1 PERFORMANCE COMPARISON BETWEEN SINGLE SCENE AND CROSS SCENE

Based on the selected three models in Figure 4, we summarize the following conclusions: (1) Across almost all abilities, model performance under the cross-scene setting is worse than under the single-scene setting. (2) Regardless of the setting, Gemini 2.5 Pro consistently outperforms Qwen2.5-VL-7B and InternVL3.5-8B across all abilities, particularly in Content Summary and Memory Recall, while also sustaining relatively high performance under the cross-scene condition. In comparison, InternVL3.5-8B performs comparably to Gemini 2.5 Pro in the single-scene setting but suffers from substantial degradation in the cross-scene setting. Meanwhile, Qwen2.5-VL-7B shows severe performance drops in Proactive Interaction and Memory Recall under cross-scene evaluation.

#### 4.3.2 PERFORMANCE OF DIFFERENT VIDEO LENGTHS

To study the impact of video length on model performance, videos are grouped into different length ranges. From Figure 5, we find that: (1) Model performance generally decreases as video length increases, suggesting that longer videos pose greater challenges for capturing and reasoning over multi-turn dialogue content. (2) Higher-capacity models, such as Gemini 2.5 Pro, tend to achieve higher overall scores across all video lengths compared to smaller models like Qwen2.5VL-7B. However, all models exhibit noticeable performance drops for very long videos. (3) The performance gap between models is more pronounced for shorter videos, while for longer videos, the performance difference narrows.

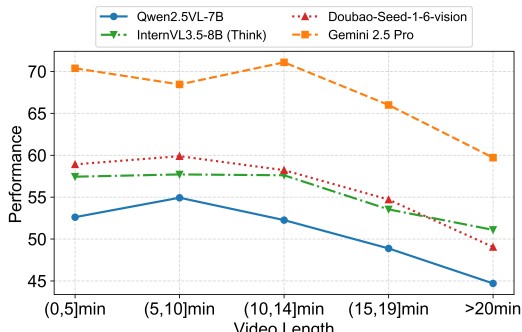

Figure 5: Performance comparison of four MLLMs across diverse video lengths.

### 4.3.3 MODEL PERFORMANCE ACROSS DIALOGUE TURNS

To evaluate the impact of dialogue length on model performance, we conduct experiments with dialogues of varying total turn numbers with Gemini-2.5-Pro and Qwen2.5VL-7B. Several key observations can be drawn from the results shown in Figure 6: Model performance tends to improve as the total number of turns increases, although the degree and stability of this improvement vary across models. For example, Gemini 2.5 Pro remains stable at 69.22 in both 5-turn and 6-turn settings, experiences a slight drop to 67.78 at 7 turns, but then reaches its peak performance of 75.56 at 8 turns. In contrast, Qwen2.5-VL-7B rises from 52.84 at 5 turns to 54.76 at 7 turns, and then achieving a significant jump to 62.92 at 8 turns. This suggests that dialogue length plays a dual role in multi-turn video understanding: on the one hand, offering more contextual cues beneficial for reasoning, while on the other hand increasing the burden of sustaining co-

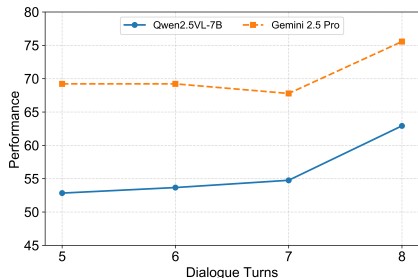

Figure 6: Performance comparison of Qwen2.5-VL-7B and Gemini 2.5 Pro across dialogue turns.

herent dialogue states. One possible reason for this pattern is larger models are generally able to integrate contextual information more efficiently, leveraging additional turns to further improve. Smaller models, on the other hand, tend to rely more heavily on the accumulation of dialogue context across multiple turns.

### 4.3.4 EFFECT OF DIFFERENT NUMBERS OF FRAMES

In Figure 7, results of Qwen2.5-VL-7B are grouped according to the number of frames, several distinct trends emerge from the results:

(1) **Topic Shifting.** The performance on topic shifting remains largely unaffected by the number of frames. This suggests that the ability to adapt to unexpected user queries and maintain coherent responses is primarily dependent on dialogue-level reasoning rather than fine-grained visual information.

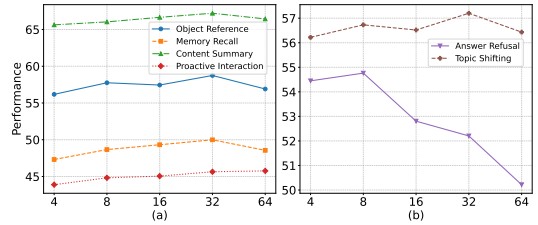

Figure 7: Ablation results of frames on different abilities. (a) Performance of object reference, memory recall, content summary, and proactive interaction; (b) Performance of answer refusal and topic shifting.

(2) **Anwser Refusal.** Models perform better on answer refusal cases when fewer frames are provided. With limited visual evidence, the model becomes more cautious in generating answers and is less likely to hallucinate unsupported content, while when more frames are provided, the model may overfit to irrelevant visual cues and produce unwarranted responses, leading to decreased performance on this ability.

(3) **Long Context Benefits.** For other four abilities, as shown in Figure 7 (a), models' performance consistently improves with more frames, because longer visual evidence provides richer contextual signals, which support more accurate reasoning.

## 5 CONCLUSION

In this paper, we presented MT-Video-Bench, a holistic benchmark for evaluating MLLMs in multi-turn video dialogues. Unlike prior video understanding benchmarks that primarily focus on single-turn factual perception, MT-Video-Bench jointly assesses perceptivity and interactivity through six carefully defined capabilities, covering tasks such as memory recall, topic shifting, and proactive interaction. Our evaluation of 20 state-of-the-art models provides insightful findings, and we hope our MT-Video-Bench can establish a rigorous foundation for future research, highlighting the need for models that can reason over long contexts while engaging in natural, adaptive conversations.

## ETHICS STATEMENT

Our work introduces MT-Video-Bench, a holistic video understanding benchmark for evaluating MLLMs in multi-turn dialogues, and does not pose direct ethical concerns. All videos and annotations are either synthetically generated or sourced from publicly available video websites, containing no personally identifiable information or sensitive content.

## REPRODUCIBILITY STATEMENT

The MT-Video-Bench dataset construction process is fully reproducible using the provided construction details and prompts. We detail the steps for acquiring, splitting, and processing video data, as well as the methods for generating multi-turn dialogues and evaluation checklists. All resources, including prompts, task definitions, and validation procedures, are made publicly available. Researchers can easily replicate our dataset generation by following the instructions.

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

# A    THE USE OF LARGE LANGUAGE MODELS (LLM)

Yes, LLMs are utilized in the preparation of this paper. Specifically, LLMs are employed to refine the writing, further enhancing the readability and quality of the paper.

# B    DETAILS ON THE DATA GENERATION

The prompts we utilize for multi-turn dialogues generation are shown below. First, we present the prompt template, followed by the specific task instructions and requirements for each of the six tasks.

---

**Prompt Template**

You are a **Video QA Orchestrator** designed to rigorously evaluate a video-understanding model's capabilities with a primary focus on # Task # within a dialogue:

1. Core Task Instructions
{Task Instructions}

2. Supporting Basic Capabilities
- Basic capabilities testing may be introduced appropriately in the dialogue to provide richer conversational context, but their purpose is to better assist in testing **object reference** capabilities.
- The questions (User) and answers (Assistant) of basic ability tests need to be detailed and rich.

A. Scene Understanding
1) Entity Recognition.  2) Event Comprehension.  3) Temporal Relations.  4) Spatial Relations. 5) OCR/Text

B. Scene Reasoning
1) Causal Inference. 2) Intent Inference. 3) Counterfactuals. 4) Spatiotemporal Reasoning. 5) Information Update. 6) Multi-Hop Logic

3. Target Task Capability
{Task Capability}

4. Input: A video clip.

5. Output:
Self-generated, cohesive multi-round dialogue (5-8 rounds) systematically probing capabilities of # Task #.

Output Format
{
"Round 1":
"User": "User question",
"Assistant": "Assistant Answer"
,
...
"Round X":
"User": "User question",
"Assistant": "Assistant Answer"

}

---

## Object Reference

**Task Instructions**

- Mandatory Contextual Dependency: Every question MUST require information from previous rounds to be answerable. Object Reference questions must rely on the content of prior rounds for context. The reference should be indirect, and the model should infer which entity or event is being referred to based on prior dialogue. In other cases, entities and events can be explicitly stated.
- Strict Anti-Isolation: Never create standalone questions. Each round must chain to the previous multi-turn dialogue through explicit or implicit references.
- Answer Grounding: All answers must derive from video content or logical inferences from it. Answers must demonstrate an understanding of prior context to provide coherent responses.
- Clarity in Pronoun Reference: When using pronouns in **Object Reference** questions, ensure that the pronoun's reference is inferred from previous dialogue, not explicitly mentioned in the current round. The model must deduce from the context which entity or event the pronoun refers to. The model should *not* assume the pronoun refers to a subject in the current round, unless explicitly clarified by the dialogue history.

**Task Capability**

- This capability of Object Reference MUST be tested **at least once** in any round after the first. **Multiple or Continued** tests are encouraged.
- For capability of Object Reference, the answer (Assistant) should correctly identify the abstract Object Reference mentioned in Question (User) (e.g., "the action mentioned," "that dog you described," "that man," etc., which refer to content from the dialogue history).
- Indirect Reference: For **Object Reference** questions, **do not explicitly state** who or what is being referenced; it should be inferred from context. The model must deduce the entity or event being referred to based on prior conversation.
- Other Abilities: In other questions where **Object Reference** is not the focus, feel free to explicitly state the entity or event being referred to.
- Pronoun Clarification in Answers: When answering questions that involve pronouns, clarify the reference by placing the referent in parentheses in the answer (e.g., "Penguin Dad (he) went to the sea"). This ensures that the pronoun reference is clear and explicitly tied to the correct entity from previous rounds.

## Memory Recall

**Task Instructions**
- Mandatory Contextual Dependency: Every question MUST require information from previous rounds to be answerable. **Memory Recall** questions must refer to detailed descriptions from any prior round. The model must recall and identify which specific round the referenced content came from. After recalling this, the model should deepen the exploration by relating it to further aspects of the video content.
- Strict Anti-Isolation: Never create standalone questions. Each round must chain to previous multi-turn dialogue through explicit or implicit references.
- Answer Grounding: All answers must derive from video content or logical inferences from it. Answers must demonstrate an understanding of prior context to provide coherent responses.
- Memory Recall: Questions (User) should reference specific content or actions described in earlier rounds (e.g., "As you mentioned earlier, the penguin hesitated.... In which round of the conversation was this mentioned? ....") and then build upon that context to ask deeper, video-based questions. The Answer (Assistant) must first identify the round from which the referenced content originated.
- No Explicit Round Numbers in Questions: **Memory Recall** questions should not explicitly mention the round number from which the content is referenced. The model must determine which round the referenced content comes from in the answer and then proceed with further exploration.

**Task Capability**
- Memory Recall capability MUST be tested **at least once** in any round after the first. **Multiple or Continued** tests are encouraged.
- For Memory Recall, the question (User) should include a detailed description from a previous round's answer (Assistant), but without specifying which round it came from. The model must first identify the round the content came from in its answer and then build upon that context to deepen the exploration.
- Further Exploration: After recalling content from prior rounds, the model should pose further questions that deepen the understanding of the video, exploring new aspects of the content.
- Other Abilities: In other questions where **Memory Recall** is not the focus, feel free to explicitly state the entity or event being referred to.

### Content Summary

**Task Instructions**
- Mandatory Contextual Dependency: Every question MUST require information from previous rounds to be answerable. **Summary** questions must rely on the content of prior rounds for context. The summary should be concise and capture the core points of the entire conversation. The model should summarize the main discussion points in the last round.
- Strict Anti-Isolation: Never create standalone questions. Each round must chain to previous multi-turn dialogue through explicit or implicit references.
- Answer Grounding: All answers must derive from video content or logical inferences from it. Answers must demonstrate an understanding of prior context to provide coherent responses.
- Clarity in Summary: In the final round (Summary), the model should summarize the key elements discussed, such as topics, events, and entities involved in the conversation. The summary should highlight the main themes of the dialogue, but without introducing new information.

**Task Capability**
- Content Summary capability MUST be tested **once** in the final round. It should consolidate the entire conversation.
- For the capability of Content Summary, the answer (Assistant) should provide a **concise** and **coherent summary** of the conversation, covering the key topics, entities, events, or actions mentioned in the previous rounds.
- Concise and Comprehensive Summary: The final summary should:
- Cover the **main events** of the conversation.
- Identify **key entities** discussed.
- Capture the **important actions** or **decisions** made throughout the conversation.
- Be **concise** but should not leave out key elements discussed.
- Other Abilities: In other rounds, feel free to ask questions that focus on specific abilities (e.g., event comprehension, object recognition, etc.). These should be used to help build up the context for the final summary.

### Answer Refusal

**Task Instructions**
- Mandatory Contextual Dependency: Every question MUST require information from the video to be answerable. **Answer Refusal** occurs when the question refers to video content that does not exist in the video. The model should identify that the content referenced in the question is missing and refuse to answer accordingly.
- Strict Anti-Isolation: Never create standalone questions. Each round must chain to previous multi-turn dialogue through explicit or implicit references, but only if the referenced content exists in the video.
- Answer Refusal Grounding: All answers must derive from video content or logical inferences from it. If the question refers to non-existent video content, the model must explicitly refuse to answer, stating that the content does not exist in the video.
- Answer Refusal: Questions (User) should reference content or actions that are expected to be present in the video. If the content referred to is non-existent in the video, the model must refuse to provide an answer, explaining that the specific content does not exist in the video.

**Task Capability**
- Answer Refusal capability MUST be tested **at least once** in any round after the first. **Multiple or Continued** refusal tests are encouraged.
- For **Answer Refusal**, the question (User) should ask about content in the video, but without specifying where the content should be found. If the content does not exist in the video, the model must refuse to answer and give some explanation.
- Further Exploration: After refusing to answer based on non-existent content, the model should seek clarification or adjust the conversation to explore available content in the video.
- Other Abilities: In other questions where **Answer Refusal** is not the focus, feel free to explicitly state the entity or event being referred to.

### Topic Shifting

**Task Instructions**
- Mandatory Contextual Dependency: Every question (User) must initially focus on video content. Topic Shifting occurs when the user asks a question or makes a statement that shifts to any other topic (video-related OR unrelated). The model should recognize the new topic and respond appropriately.
- Topic Shifting Recognition: If a question (User) shifts to a new topic (including non-video topics), the model must identify this shift and provide an appropriate answer.
- Adapting to New Topics: When the model recognizes a topic shift, it must adjust its response to the new topic while ignoring obsolete context.

**Task Capability**
- Topic Shifting capability MUST be tested at least twice through shifts to non-video topics**, with multiple back-and-forth transitions between video and non-video topics.
- Pattern Example: Video topic → Non-video topic → Video topic → Non-video topic → etc.
- Minimum Video Content: At least two rounds must focus exclusively on video content.
- For Topic Shifting, the model must:
1. Identify shifts to any new topic (video-related or unrelated)
2. Maintain response stability during repeated topic transitions
3. Seamlessly resume video discussions after non-video tangents

---

**Proactive Interaction**

**Task Instructions**
- Context-Driven Engagement: Each Bot response must be grounded in the video's content and must explicitly or implicitly invite the User to stay engaged.
- Never Isolated: Bot replies must reference the video and maintain dialogue continuity. Standalone statements are not allowed.
- Proactive Engagement Strategy: When user responses are neutral or show low engagement, the Bot must actively reignite curiosity by asking video-specific, open-ended, thought-provoking questions.
- Content-Guided Curiosity: Bot questions should be deeply tied to the video — e.g., character motivation, visual detail interpretation, or causal/temporal implications — to pull the user back into the conversation.
- No rhetorical or vague questions like "Don't you think?" Instead, ask clear, content-specific, curiosity-prompting questions (e.g., "Why do you think the character hesitated before opening the door?").

**Task Capability**
- From **Round 2 onward**, the **Bot must trigger Proactive Interaction** when the user's input is neutral or disinterested.
- Bot should:
- Highlight overlooked details.
- Ask novel or deeper questions.
- Offer an unexpected angle of interpretation. - Questions should **always tie to specific visual or narrative elements** in the video and encourage user response.

---

# C   DETAILS ON THE EVALUATION SETTINGS

## C.1   FRAMES AND RESOLUTION

The experimental details regarding the number of frames and resolution are provided as follows:

- For most models, we uniformly sample 32 frames. Each frame is resized such that its longer side is limited to 720 pixels, with the shorter side scaled proportionally. The following models adopt this setting: Gemini series (Team, 2025), Doubao-Seed-1.6-vision (Seed, 2025), Qwen2.5 series (Bai et al., 2025), LLaVA-Video series (Zhang et al., 2024d), LLaVA-Onevision series (Li et al., 2024b), LLaVA-NeXT-Video series (Zhang et al., 2024c), MiniCPM series (Yao et al., 2024), VideoChat-Flash series (Li et al., 2024e), VideoLLaMA3 series (Zhang et al., 2025a).

- For the InternVL3.5 series (Wang et al., 2025b), we sample 32 frames and set the resolution to $448 \times 448$, following the model-specific requirements.

- For the InternVideo2.5 series (Wang et al., 2025d), we sample 32 frames and set the resolution to $728 \times 728$, in order to minimize the impact of varying experimental settings.

## C.2   PROMPT FOR INFERENCE

For most models, we use the following unified inference prompt:

---

**Inference prompt**

You are an AI assistant designed to answer questions about the video. Here are the previous conversations:
{conversational history}
Now, answer the following question, taking into account the conversation history:
{question}

---

For the think mode of InternVL3.5 series, we incorporate the system prompt provided by the official documentation.

## C.3 PROMPT FOR GENERATING EVALUATION CHECKLISTS

In this section, we provide the prompts for generating evaluation checklists of six tasks in MT-Video-Bench. First, we present the prompt template, followed by the definitions and requirements for each of the six tasks.

---

**Prompt Template**

You are an expert in evaluating AI video dialogue models. Your task is to create exactly 5 highly specific yes/no questions based on the provided user query and ground-truth answer to assess the model # Ability # in the conversation across multiple turns.

# Ability Definition #

# Requirements#

**Inputs**:
- User Question: {question}
- Ground Truth Answer: {answer}

**Output Format**:
Generate a valid JSON object only:

```
{
"Q1": "[Concrete question about specific ability]",
"A1": "[Yes/No]",
"Q2": "[Concrete question about specific ability]",
"A2": "[Yes/No]",
"Q3": "[Concrete question about specific ability]",
"A3": "[Yes/No]",
"Q4": "[Concrete question about specific ability]",
"A4": "[Yes/No]",
"Q5": "[Concrete question about specific ability]",
"A5": "[Yes/No]"
}
```

---

**Object Reference**

**Ability Definition**: Object Reference
This ability involves the model's capacity to:
- Accurately identify and reference specific entities (such as objects, people, locations, or events) mentioned in the dialogue.
- Resolve pronouns (e.g., "it", "them", "that one") and implicit references correctly based on the context.
- Maintain coherence in references across multiple rounds of dialogue, ensuring the model understands and correctly links these references.

**Requirements**:
1. Generate exactly 5 yes/no questions
2. Ensure answer balance (mix of Yes/No responses)
3. Create highly concrete questions that:
- Check if the model resolves pronouns or implicit references correctly (1-2 questions).
- Verify that the model's answer is correct (3-4 questions).
- Example of GOOD question: "Does the model accurately resolve the pronoun 'it' to refer to the 'book on the table' mentioned before?"
- Example of BAD question: "Does the model remember what was said earlier?"

## Memory Recall

**Ability Definition**: Memory Recall

This ability involves precisely extracting and referencing specific information mentioned earlier in multi-turn conversations, including:

- Accurately retrieve dialogue history, clearly identifying that the current content was mentioned in round xx
- Clearly determine whether the model correctly understands and references content from previous rounds of dialogue
- Check if the model's subsequent answer provides an accurate exploration based on this content

**Requirements**:

1. Generate exactly 5 yes/no questions
2. Ensure answer balance (mix of Yes/No responses)
3. Create highly concrete questions that:
- Explicitly assess whether the model accurately locates and references content from a specific round in the dialogue history
- Check whether the model's subsequent answer is accurate based on the historical content and video
- Avoid abstract or generalized phrasing
- Example of GOOD question: "Does the model accurately state that 'the cat sleeping on the sofa was discussed in round 2'?"
- Example of BAD question: "Does the model remember the previous dialogue content?"

## Content Summary

**Ability Definition**: Content Summary

This ability involves accurately summarizing the main characters, event developments, and key details presented in the video in combination with the user's historical dialogue, while:

- Clearly presenting the core content, main characters, and important outcomes shown in the video and mentioned in the dialogue
- Accurately organizing the video content according to the actual sequence of events, and aligning it with the progression of the dialogue
- Appropriately supplementing the dialogue summary with relevant details from the video, ensuring coherence and completeness
- Avoiding the inclusion of information not present in either the video or the dialogue, as well as overly trivial details

**Requirements**:

1. Generate exactly 5 yes/no questions
2. Ensure answer balance (mix of Yes/No responses)
3. Create highly concrete questions that:
- Focus on critical characters, objects, locations, and the final results that appear in the video and/or are discussed in the dialogue
- Emphasize the event sequence as reflected in both the video content and the dialogue flow
- Appropriately check whether the model supplements the dialogue with relevant video details and aligns video events with dialogue progression
- Avoid abstract or generalized phrasing

## Answer Refusal

**Ability Definition**: Answer Refusal
This ability involves appropriately refusing to answer user questions about events or details that are not present in the video content, while:
- Providing a respectful explanation that references specific video limitations
- Clearly stating that particular inquired events/details were not shown in the video
- Avoiding fabrication of specific information not supported by the video evidence

**Requirements**:
1. Generate exactly 5 yes/no questions
2. Ensure answer balance (mix of Yes/No responses)
3. Create highly concrete questions that:
- Directly reference specific elements mentioned in the user question
- Focus on particular video content that was or wasn't shown
- Avoid abstract or generalized phrasing
- Example of GOOD question: "Does the model explicitly state that the video does not show a red car crashing into the tree?"
- Example of BAD question: "Does the model correctly identify irrelevant content?"

## Topic Shifting

**Ability Definition**: Topic Shifting
This ability involves effectively managing unpredictable topic switches during conversations, including:
- Maintaining contextual awareness of both previous and new topics
- Seamlessly integrating the new topic or switching back to the original topic while respecting conversation flow, and ensuring responses stay focused and do not introduce information unrelated to the current topic
- Accurately answering the current topic query with relevant and appropriate information

**Requirements**:
1. Generate exactly 5 yes/no questions
2. Ensure answer balance (mix of Yes/No responses)
3. Create highly concrete questions that:
- Focus on the model's response, specifically assessing both the **accuracy of its answers** and **whether its content remains strictly relevant to the topic at hand**
- **Directly reference the key facts, entities, or specific information** present in the dialogue, rather than general conversational cues or transition phrases.
- Target precise transition moments and evaluate if the response contains any off-topic or extraneous information
- For questions where the expected answer is "Yes": Avoid abstract or generalized phrasing, focus on specific verifiable facts from the response. For questions where the expected answer is "No": Do not fabricate specific details that are not present in the model's actual answer
- Example of GOOD question: "Does the model accurately state that 'The Grove' closes at 11 PM on weekends?"
- Example of GOOD question: "Does the model introduce unrelated facts about the restaurant's ownership history?"
- Example of BAD question: "Does the model use good transition words when switching topics?"

Proactive Interaction

**Ability Definition**: Proactive Interaction
This ability includes the model proactively asking the user novel or more in-depth questions, or offering other discussion angles based on the video content to encourage user responses. To evaluate this ability, the following aspects should be considered:
- The AI assistant's accuracy in restating the topics mentioned by the user and in providing statements about the video content.
- The AI assistant's initiative in contributing beyond direct answers, including asking relevant follow-up questions, proposing new topics, and maintaining the natural flow of conversation by encouraging the user to share more information or thoughts.
- The AI assistant's balance between proactivity and responsiveness, ensuring its interactive elements are appropriate, engaging, and adaptive to the user's input without derailing the conversation.

**Requirements**:
1. Generate exactly 5 yes/no questions
2. Ensure answer balance (mix of Yes/No responses)
3. Create highly concrete questions that:
- Check the accuracy of the model's statements about the topics mentioned by the user
- Explicitly assess whether the model proactively asks a novel or more in-depth question based on the video content
- Evaluate if the model's interactive attempts appropriately maintain conversation flow and engagement
- Design all questions strictly based on the elements present in the user question and the ground truth answer. Do not introduce new characters, objects, or events that are not mentioned. Negative (No) cases should focus on whether the model fails to show proactive interaction, rather than expecting it to cover additional directions not included in the ground truth.

## C.4 PROMPT FOR EVALUATION BASED ON CHECKLISTS

Evaluation Prompt

Given a current user question, a standard answer, a model-generated answer, and a checklist of several questions, generate answers to the questions in the checklist to evaluate the model's performance.

# User Question:
{question}

# Standard Answer:
{standard answer}

# Model Answer:
{model answer}

# Checklist:
{checklist}

Output Format:
Just output yes or no, do not generate redundant content.

## D MORE CASES

We present some failure cases, and in both the ground truth and the model's response, the portions highlighted in red. In the checklist, the content inside square brackets denotes the correct answer.

Answer Refusal

**User**: Describe the meal the man is eating in the video clip from 00:06:22 to 00:07:00. What are the different components on his plate, and what does he say about the texture and taste of the meatballs when he first tries them?

**Assistant**: The man is eating a meal consisting of several components on a white plate: meatballs covered in a creamy brown sauce, ...

**User**: What kind of bread does he eat to balance the saltiness mentioned in the video clip from 00:06:22 to 00:07:00?

**Assistant**: I'm sorry, but the video does not show the man eating any bread with his meal. His plate contains meatballs, mashed potatoes, salad, and jam, but no bread is visible or mentioned.

**User**: You're right, there's no bread. Instead of bread, what does he add to the meatballs later, and how does he say this addition changes the flavor of the dish?

**Assistant**: He later dips a meatball into the dark red jam. He explains that ...

**User**: The man purchases a drink to go with his meal. What brand is the drink, and according to the video, what is its price in euros and its equivalent in Chinese yuan?

**Assistant**: The man purchases a bottle of Coca-Cola. A shot of the drink cooler shows a price tag of '3.30'. The man confirms this ...

**User**: In the video clip from 00:06:22 to 00:07:00, what brand of chips or snacks are being sold next to the Fanta bottles?

**Assistant**: I cannot answer that question as the video does not show any chips or snacks for sale in the drink cooler ...

**User**: At the end of the video, the camera shows the view from the train window. What does the scenery outside look like, and what time of day is it?

**Assistant**: The scenery outside the train window at the end of the video is a snowy landscape. It appears to be nighttime, ...

# User Question:
Regarding the people sitting in the background of the dining car in the video clip from 00:06:22 to 00:07:00, what are the other passengers eating and drinking?

# Standard Answer:
I'm sorry, but the video does not provide a clear view of what the other passengers in the background are eating or drinking. While other people are visible, the camera remains focused on the main speaker and his meal, and the details of other passengers' tables are out of focus and not shown.

# LLaVa-Next-Video-7B's Answer:
The other passengers in the background of the dining car are eating meatballs with mashed potatoes and salad, and drinking Coca-Cola.

# Checklist:
Q1: "Does the model explicitly state that the video does not provide a clear view of what the other passengers in the background are eating or drinking?
A1: No[Yes].
Q2: Does the model explain that the camera remains focused on the main speaker and his meal as a reason for the inability to provide the requested details?
A2: No[Yes].
Q3: Does the model state that the details of other passengers' tables are out of focus?
A3: No[Yes].
Q4: Does the model deny that any other passengers are visible in the background of the dining car?
A4: No[Yes].

**Object Reference**

**User**: What is the large geographical feature called that is introduced on the map at the beginning of the video clip from 00:02:40 to 00:03:42, what is its approximate area, and what comparison is made about its potential size?

**Assistant**: The geographical feature is the Hengduan Mountain system. The on-screen text states its area is over 600,000 square kilometers. The video makes a comparison that if all its folds were flattened, it would expand to over three times its current area, large enough to cover the entire Central China region to the south.

**User**: Following the size comparison, the video provides specific linear dimensions for the Hengduan Mountain system. What are its east-west and north-south spans?

**Assistant**: According to the text displayed on the map, the Hengduan Mountain system spans over 800 kilometers from east to west and over 1300 kilometers from north to south.

**User**: The video then shows a series of beautiful natural landscapes. Considering the context from our previous discussion, what are the different types of scenery shown within that vast mountain system?

**Assistant**: Within the Hengduan Mountain system (the mountain system previously discussed), the video showcases a variety of scenery, including vast, winding, snow-capped peaks against a blue sky, lush green valleys with rivers flowing through them, autumn forests with yellow trees, turbulent, muddy rivers rushing through steep gorges, and terraced turquoise pools of water surrounded by frosted trees.

**User**: After showing the natural landscapes, the video shifts focus to people. What are the people shown doing in these environments?

**Assistant**: The video shows people, equipped with backpacks, hiking and trekking through the landscapes. One person is seen walking up a grassy, sunlit hill with mountains in the background. Another is shown standing alone on the edge of a calm, misty lake, reflecting the mountains. A third person is silhouetted on a high ridge, looking out at a massive snow-covered peak.

# User Question:
What does the on-screen text say about their attitude or experience during these difficult journeys?

# Standard Answer:
The text suggests that despite the challenges, they (the hikers) are resilient and forward-looking. It states, "Regardless of the high mountains and long roads, though there's mud on our clothes, we always see thousands of miles of rosy clouds before our eyes."

# InternVideo2.5-8B's Answer:
The on-screen text says "However, the hardships of life are not the end of the world."

# Checklist:
Q1: Does the model accurately resolve the pronoun 'It' in the phrase 'It states' within the answer to specifically refer to 'The text' mentioned earlier?
A1: Yes[Yes].
Q2: Does the model correctly infer that 'their' in the user's question refers to the specific subjects identified as 'they (the hikers)' in the answer?
A2: No[Yes].
Q3: Does the model explicitly label the 'high mountains and long roads' mentioned in the quoted text as 'difficult journeys' to directly address the user's query?
A3: No[No].

### Topic Shifting

**User**: What animals can be seen in the opening scenes of the video clip from 00:00:00 to 00:00:23, up to the 00:00:16 mark?

**Assistant**: In the opening scenes, a small black cat is shown walking on a porch at the beginning of the video, and a brown chicken is seen walking on a grassy lawn around the 00:00:15 mark.

# User Question:
Speaking of cats, I'm thinking of getting one. What are some key differences in temperament between a Siamese cat and a Maine Coon?

# Standard Answer:
That's a great question, and it's a shift from the video content. In general, Siamese cats are known for being very vocal, intelligent, and social, often forming strong bonds with a single person. They are typically very active and require a lot of stimulation. Maine Coons, on the other hand, are often called 'gentle giants.' They are known for their friendly, good-natured, and playful personalities but are generally less demanding and vocal than Siamese cats. They tend to be more independent while still enjoying human companionship.

# Qwen2.5-VL-3B's Answer:
The video does not contain any information about Siamese or Maine Coon cats or their temperaments. It appears to be focused on gardening and cooking activities.

# Checklist:

Q1: Does the response accurately state that Siamese cats are known for being very vocal?
A1: No[Yes].
Q2: Does the response describe Maine Coons as generally less demanding than Siamese cats?
A2: No[Yes].
Q3: "Does the response provide historical information about the origin of either cat breed?
A3: No[Yes].

### Proactive Interaction

**User**: The video clip from 00:00:45 to 00:02:57 starts by showing two different video editing software interfaces side-by-side on two laptops.

**Assistant**: Indeed, we see both Premiere Pro and DaVinci Resolve being used. The interfaces look quite complex. Based on that initial comparison, what specific design elements in either interface first catch your eye as being potentially more user-friendly or powerful?

# User Question:
They both just look like typical editing software.

# Standard Answer:
I can see that. However, at 00:01:06, the video zooms in on a new purple tool icon in the Premiere Pro interface labeled 'AI New Function'. This seems to be a significant addition. What kind of groundbreaking capability do you think an 'AI Generative Extension Tool' might provide an editor that traditional tools can't?

#Qwen2.5-VL-7B's Answer:
The video clip continues to showcase various aspects of video editing software, including the use of Premiere Pro and DaVinci Resolve. It highlights the complexity and features of these tools, such as the timeline, audio editing, and color grading. The video also emphasizes the importance of having the right hardware for optimal performance, mentioning the need for large graphics cards, ample memory, and solid-state drives. Additionally, it touches on the cost-effectiveness of using these software solutions, suggesting that they can be a good investment for professionals in the industry.

# Checklist:
Q1: Does the system accurately acknowledge the user's initial observation about the software looking like typical editing software?
A1: Yes[Yes].
Q2: "Does the system proactively introduce a specific, novel detail from the video related to a new AI functionality?
A2: No[Yes].
Q3: Does the system ask a specific, open-ended question that encourages the user to consider the future implications or capabilities of the identified AI feature?
A3: No[Yes].
Q4: Does the system immediately offer multiple distinct avenues for discussion or ask more than one primary follow-up question in its response?
A4: Yes[Yes].
Q5: Does the system provide a specific answer or interpretation of the AI feature's capabilities before asking the user for their thoughts?
A5: No[No].

Content Summary

**User**: Based on the video clip from 00:03:58 to 00:06:04, answer the following question. Who is the man in the dark blazer, and what historical detail does he provide about the door he is standing next to?"

**Assistant**: The man appears to be a representative or manager of a historic establishment. He explains that ...

**User**: After discussing the door from 1873, what other piece of furniture does the man point out, and what is its historical significance?

**Assistant**: He then gestures to a chair and states that the chairs in the establishment ...

**User**: The man then moves to a newspaper rack. Based on his explanation, why are physical newspapers considered an essential, 'part and parcel' element of this coffeehouse?

**Assistant**: He explains that having a selection of newspapers, including ...

**User**: Considering his emphasis on the tradition of physical newspapers, what modern challenge does he acknowledge, and how does the coffeehouse's practice of sharing papers reinforce its traditional values?

**Assistant**: He acknowledges that they are now competing with online media. The practice of reading a physical paper and then ...

**User**: Finally, the man is seen seated while discussing an 'unwritten law.' What is this law, and according to another patron, what core purpose of the coffeehouse does it serve?

**Assistant**: The unwritten law is that patrons face no pressure to keep ordering and ...

# User Question:
Based on our entire conversation, please summarize the man's role and the key traditions of the Viennese coffeehouse that he has presented.

# Standard Answer:
The man in the dark blazer appears to be a representative or manager of a historic Viennese coffeehouse. He emphasizes several key traditions and aspects that define this establishment: 1. The historical significance of its furnishings, such as chairs used for over 120 years and doors from 1873, highlighting the long-standing heritage of the place. 2. The tradition of having physical newspapers available for patrons, including international editions, which is an integral part of the coffeehouse's cultural identity and provides a communal activity. 3. An unwritten law stating that there is no pressure on patrons to keep ordering food and drinks; they can stay as long as they like, reinforcing the primary purpose of social interaction and leisure within the coffeehouse environment. These elements collectively underscore the unique atmosphere and longstanding customs that make the Vienna coffeehouse experience distinct and cherished by both locals and visitors.

# MiniCOM-o's Answer:
The man, acting as a guide, highlights the deep-rooted traditions of the Viennese coffeehouse. He showcases the historical longevity of its fixtures, such as the door from 1873 and 120-year-old chairs. He emphasizes the cultural importance of providing physical newspapers for communal reading, contrasting this tradition with modern online media. Finally, he explains the coffeehouse's core philosophy through its 'unwritten law,' which allows patrons to stay indefinitely, fostering a unique environment for unhurried social interaction rather than just consumption.

# Checklist:
A1: Does the summary specify the man's formal title, such as a manager or representative of the establishment?
A2: No[Yes].
Q2: Does the summary explain the 'unwritten law' as permitting patrons to stay indefinitely for social interaction rather than solely for consumption?
A2: Yes[Yes].
Q3: Does the summary highlight the historical significance of the coffeehouse's furnishings?
A3: Yes[Yes].

## Memory Recll

**User**: Describe the initial interaction between the two octopuses at the beginning of the video clip from 00:04:23 to 00:06:12. What are they doing, and what objects are involved in their interaction?

**Assistant**: The video opens with a reddish-orange octopus on the sandy seabed, gently extending a tentacle to touch another octopus that is partially sheltered inside a dark, mossy coconut shell. This initial scene suggests a curious or friendly interaction between the two, establishing a bond.

**User**: Following this initial friendly interaction, what causes the octopus inside the coconut shell to leave its shelter?

**Assistant**: The octopus inside the shell, which is referred to as the 'coconut octopus', leaves its shelter because it needs to find food. The narration explains that it must abandon its protection to go foraging, which sets up the subsequent events of the video.

# User Question:
You mentioned that the coconut octopus had to leave its shelter to forage. Can you recall when this detail was discussed, and what was the immediate consequence for the other octopus after its friend left?

# Standard Answer:
This was discussed in Round 2. The immediate consequence for the other octopus was that it became extremely vulnerable. Without the protection of a shell, it was caught by the ocean current and began to drift helplessly across the seabed, unable to control its movement.

# VideoChat-Flash-7B's Answer:
The detail about the coconut octopus needing to forage and leaving its shelter was discussed after the initial friendly interaction between the two octopuses. The immediate consequence for the other octopus, which is the reddish-orange one, was that it had to fend off a shark alone.

# Checklist:
Q1: Does the model accurately identify 'extremely vulnerable' as the immediate consequence for the other octopus after its friend left?
A1: No[Yes].
Q2: Does the model accurately explain that the other octopus was caught by the ocean current after its friend left?
A2: No[Yes].
Q3: "Does the model claim that the other octopus immediately managed to find a new protective shell after its friend left?
A3: No[No].

