# OpenReview forum: "MT-Video-Bench:  A Holistic Video Understanding Benchmark for Evaluating Multimodal LLMs in Multi-Turn Dialogues"
_ICLR.cc/2026/Conference — ICLR 2026 Conference Withdrawn Submission_

### Official Review · Reviewer_qhiz · 2025-10-30

**Soundness:** 3
**Presentation:** 3
**Contribution:** 2
**Rating:** 4
**Confidence:** 4

**Summary:**

In this work, the authors propose a MT-Video-Bench, a holistic video understanding benchmark for evaluating MLLMs in multi-turn dialogues. Specifically, they carefully collect and annotate 987 multi-turn dialogues in six core capacities of perceptivity and interactivity. Finally, they evaluate various state-of-the-art MLLMs to show the potential of proposed benchmark.

**Strengths:**

* Clarity

The paper is well-written with good structure. Hence, the clarity is basically good.

* Significance

This paper focuses on evaluating multi-turn video dialogue capacity of MLLMs, which is an important and practical problem for video understanding. Hence, the significance is basically OK for video research community.

**Weaknesses:**

* Capacity Evaluation

In this work, the authors propose to evaluate six core capabilities of perceptivity (object reference, memory recall, content summary) and
interactivity (answer refusal, topic shifting, proactive interaction). However, the reason why to choose these tasks or capacities is not well explained. I do understand that, in general, it is necessary to evaluate perceptivity and interactivity, especically for long videos.

1) Then, why to choose these six tasks? It that sufficient to evaluate both perceptivity and interactivity? For example, in the perception test benchmark [ arXiv:2305.13786], the tasks are designed, inspired by human perception screening tests in developmental psychology or medicine.

2) Please show various capacities by visualizing more videos. It makes these capacities more understandable.

* Reference

The recent work [VRBench: A Benchmark for Multi-Step Reasoning in Long Narrative Videos, ICCV 2025] proposes the similar topic for video understanding. Please clarify the key difference.

* Method Insight

It woule be more interesting to investigate or indicate how to design MLLMs to tackle the tasks in this benchmark.

* Small Size

The authors collected only 135 original videos from online platforms. The small number of videos would restrict the generalization of this  benchmark.

**Questions:**

Please see the weakness section.

---

### Official Review · Reviewer_Wc4u · 2025-10-31

**Soundness:** 3
**Presentation:** 3
**Contribution:** 2
**Rating:** 2
**Confidence:** 5

**Summary:**

This paper introduces MT-Video-Bench, a benchmark designed to evaluate Multimodal Large Language Models (MLLMs) in multi-turn video-grounded dialogues. The benchmark spans six core capabilities—three perceptual (Object Reference, Memory Recall, Content Summary) and three interactive (Answer Refusal, Topic Shifting, Proactive Interaction)—comprising 987 dialogues across 135 videos. The authors conduct extensive evaluations of 20 open-source and closed-source MLLMs, revealing significant performance gaps and challenges, particularly in cross-scene reasoning and interactive tasks. While the benchmark addresses an important gap in video-dialogue evaluation, the paper suffers from methodological omissions, insufficient ablation studies, and ambiguities in novelty claims, which undermine its overall contribution.

**Strengths:**

1. The paper introduces MT-Video-Bench which covers diverse video categories and tasks, with balanced data distribution (Figure 3).
2. Extensive experiments on 20 models provide a holistic performance analysis, highlighting limitations.

**Weaknesses:**

1. In Figure 2, The data construction lacks details on human annotation guidelines.
2. Reliance on Gemini-based checklists risks circularity, as the top-performing model (Gemini 2.5 Pro) may align better with Gemini-generated metrics.
3. The metric of accuracy (ACC) based on binary yes/no questions oversimplifies the nuanced requirements of multi-turn dialogues. For example, tasks like "Proactive Interaction" involve open-ended reasoning but are reduced to checklist-based scoring, which may not capture conversational fluency or coherence. And the paper does not report inter-annotator agreement for human validation of checklists, undermining the reliability of the evaluation protocol.
4. The uniform sampling of 32 frames (Section 4.1) may not optimally represent long videos (e.g., those exceeding 15 minutes in Figure 3c).  The paper does not justify this choice or explore adaptive sampling methods, potentially disadvantaging models designed for long-context processing.
5. Models like InternVL3.5 and InternVideo2.5 use different resolutions (448×448 vs. 728×728), but the impact of resolution variations on performance is not analyzed.  This inconsistency introduces confounding factors in comparisons.
6. The contribution and completeness are insufficient; it would be better if there were a simple, adaptable baseline that could partially address the benchmark challenge.

**Questions:**

1. Were ablation studies conducted to compare Gemini-generated metrics with those from other LLMs (e.g., GPT-4o) or human? If not, how can the authors rule out model-specific bias?
2. Did the authors explore other sampling strategies?
3. How can fair comparisons be made when resolution varies across models?
4. Is there a plan to open source the code? This is very important for reproducing the results of the paper.
5. Although the paper claims that its method significantly improves performance, it lacks sufficient experimental data and significance analysis to support this conclusion, and the results in Table 1 show only a slight improvement.
6. Could the authors provide more visualization samples?
7. Given the complexity of the benchmark, will the code for data generation, evaluation checklists, and model inference be open-sourced?  This is critical for community adoption and validation.

---

### Official Review · Reviewer_ujFi · 2025-10-31

**Soundness:** 1
**Presentation:** 3
**Contribution:** 2
**Rating:** 4
**Confidence:** 5

**Summary:**

This paper proposes MT-Video-Bench, a holistic video understanding benchmark for evaluating MLLMs in multi-turn dialogues. The benchmark comprises 987 carefully curated dialogues across 135 videos. Based on MT-Video-Bench, the authors provide a detailed evaluation of both open-source and closed-source models, highlighting the current limitations and performance discrepancies in different abilities.

**Strengths:**

1. The authors propose a new video understanding benchmark for evaluating MLLMs in multi-turn dialogues.
2. Various open-source and close-source models are evaluated.

**Weaknesses:**

1. The comparison to existing video understanding benchmarks is problematic. For example, SVBench has 49,979 QAs while the authors claim that SVBench only has 7,374 QAs in Table 1. Moreover, the authors define long videos as videos of more than 10 min. However, the sample only has 5.88 turns in dialogue. Such a number of turns cannot fully leverage information in a "long video". Moreover, while SVBench has cross-scene dialogues, the authors further define "cross-scene" as more than scenes. That's not a good definition as "cross-scene" is already defined in English.
2. The novelty of this benchmark compared to SVBench is weak. SVBench is also a video understanding benchmark for evaluating MLLMs in multi-turn dialogues, even with far more QAs and videos.
3. Important information about data is missing, e.g., the source of the videos and the distribution of video categories.
4. The evaluation is inappropriate. While the average video length is greater than 10 min, the authors only resample 32 frames of every video. Most video information is lost in the input.
5. The evaluation will be unavailable to reproduce in the future. Since Google continually deprecates the API of old Gemini models, Gemini 2.5 Flash used in the evaluation will be unavailable in the future.
6. This benchmark uses only Gemini 2.5 Flash to generate video descriptions and dialogues, which can bring model bias.

**Questions:**

Please reply to Weaknesses.

---

### Official Review · Reviewer_Gg8Y · 2025-10-31

**Soundness:** 1
**Presentation:** 2
**Contribution:** 2
**Rating:** 4
**Confidence:** 5

**Summary:**

This paper proposes a new video understanding benchmark with multi-turn dialogues for MLLMs. The benchmark mainly assesses six core competencies that focus on perceptivity and interactivity, encompassing 987 meticulously curated multi-turn dialogues from diverse domains. With MT-Video-Bench, the authors evaluate various state-of-the-art open-source and closed-source MLLMs, revealing their significant performance discrepancies and limitations in handling multi-turn video dialogues.

**Strengths:**

1.	The proposed benchmark contains 987 meticulously curated multi-turn dialogues from diverse domains, which assesses capabilities rigorously aligned with real-world applications, such as interactive sports analysis and multi-turn video-based intelligent tutoring.
2.	The experimental results reveal that current MLLMs are still weak in multi-turn dialogues for videos.
3.	Various MLLMs with different sizes are evaluated in experiments.

**Weaknesses:**

1.	This paper excessively downplays the advantages of SVBench and overstates the novelty of the proposed benchmark. SVBench already contains multi-turn dialogues for videos across multiple scenes. Moreover, the information of SVBench in Table 1 is inconsistent with which in the paper of SVBench.
2.	The source and language of the collected data is not provided.
3.	The videos are overly compressed in the evaluation. Only 32 frames of each video is input into the evaluated MLLMs, while the authors claim that videos in their benchmark have a average length of 10 minutes.
4.	The impacts of video resampling strategies should be investigated.
5.	The results of human is not provided in experiments, which is important for assessing the abilities of MLLMs.
6.	This benchmark only adopts Gemini 2.5 Flash in data annotation and evaluation. The trustworthiness of employing this model instead of other MLLMs and humans should be investigated.
7.	The authors do not provide new methods for video understanding in multi-turn dialogues.

**Questions:**

1.	What is the novelty of the proposed benchmark compared to SVBench?
2.	What’s the source and language of your videos?
3.	What are your criteria of video sampling in evaluation?
4.	How well can humans perform on your benchmark?
5.	Is Gemini 2.5 Flash reliable in data annotation and evaluation?

---

### Note · Authors · 2025-11-12

I have read and agree with the venue's withdrawal policy on behalf of myself and my co-authors.